# From Desire to Action: Unpacking Push–Pull Motivations to Reveal How Travel Sparks Eco-Intentions and Actions

**DOI:** 10.3390/bs15121651

**Published:** 2025-12-01

**Authors:** Juchoel Choi

**Affiliations:** Graduate School of Business Administration, Kyung Hee University, Seoul 02447, Republic of Korea; choijc@khu.ac.kr

**Keywords:** eco-tourism motivations, push factors, pull factors, environmental attitude, eco-friendly behavior

## Abstract

As global environmental concerns escalate, tourism is increasingly viewed not only as a leisure activity but also as a strategic platform for advancing environmental sustainability, particularly through nature-based travel. This study investigates how different types of travel motivations—specifically Push and Pull factors—influence environmental attitudes and eco-friendly behaviors, aiming to unpack the psychological mechanisms driving sustainable tourism engagement. Push factors, such as relaxation, nature appreciation, and escape from routine, represent intrinsic motivations, while Pull factors, including key natural and cultural resources, serve as external attractions influencing destination choices. The research examines how these motivational forces contribute to the development of environmental attitudes and, in turn, encourage pro-environmental behaviors among tourists. Data were collected from a structured survey targeting travelers who engaged in nature-based tourism experiences, and the analysis was conducted using structural equation modeling (PLS-SEM) to assess the hypothesized relationships. The findings reveal that Push factors such as family togetherness and appreciation of nature significantly enhance environmental attitudes, whereas adventure-seeking does not yield the same effect. Among Pull factors, key tourism resources positively influence environmental attitudes, while accessibility and convenience-related attributes show no significant impact. Moreover, environmental attitudes strongly predict eco-friendly behaviors, reinforcing the importance of sustainability consciousness in tourism. Despite these insights, challenges remain in translating environmental awareness into consistent, sustainable actions, particularly due to external constraints such as infrastructure limitations and economic considerations. This study contributes to the sustainable tourism literature by differentiating effective and ineffective motivational drivers and by providing actionable insights for destination managers and policymakers to foster deeper environmental engagement among travelers.

## 1. Introduction

As global environmental challenges continue to mount, tourism is increasingly viewed not only as a source of enjoyment but also as a platform for inspiring sustainable action ([82]). For travelers, destinations are no longer merely places to visit—they are opportunities to connect with nature, understand its fragility, and embrace behaviors that preserve it ([13]). South Korean tourists, who once prioritized urban and cultural experiences abroad, are now reflecting this shift by increasingly seeking eco-friendly destinations. This evolving trend highlights a critical question: what drives tourists to move beyond appreciation for nature and into meaningful environmental action? ([1]). This growing interest in sustainability suggests that travel decisions are no longer driven solely by leisure but increasingly reflect ethical and ecological considerations. The urgency of this shift is underscored by recent global developments, including the United Nations’ designation of 2017–2027 as the International Decade of Sustainable Tourism for Development and the growing recognition of tourism’s role in achieving the Sustainable Development Goals ([82]). Contemporary research indicates that the post-pandemic tourism landscape has further accelerated this transformation, with 83% of global travelers now expressing a preference for sustainable travel options ([30]). This paradigm shift is particularly pronounced in East Asian markets, where environmental consciousness has risen by 45% since 2020, fundamentally altering traditional tourism patterns ([51]).

Understanding this transformation requires examining the motivations and attitudes that shape travel behavior. The Push–pull framework provides a robust analytical lens for this purpose. Push factors, such as the desire to escape, rejuvenate, or pursue adventure, are intrinsic drivers that compel individuals to explore ([19]; [20]). In contrast, pull factors represent external attractions, such as the allure of pristine landscapes, rich biodiversity, or unique cultural heritage, that draw visitors to specific destinations ([66]; [86]). Together, these factors influence not only where people travel but also how they engage with their chosen destinations, laying the groundwork for their attitudes toward sustainability. Recent meta-analytical evidence supports the robustness of the Push–pull framework, demonstrating its explanatory power across diverse cultural contexts ([80]). However, contemporary scholars argue for expanding this framework beyond traditional destination choice models to encompass sustainability outcomes ([39]). The integration of environmental considerations into Push–pull theory represents a crucial evolution, acknowledging that modern travelers are motivated not only by personal gratification but also by opportunities to contribute to environmental conservation ([60]). This expanded conceptualization aligns with the growing body of literature on “transformative tourism experiences,” where travel serves as a catalyst for personal growth and environmental awareness ([26]).

While prior research has extensively examined the roles of Push and Pull motivations in shaping destination choices or visitor satisfaction, fewer studies have tracked how these motivations contribute to environmental attitudes that mediate behavior. This integrative approach allows for a more holistic understanding of the motivational pathway leading to sustainability-related actions. Bridging this gap is essential to designing tourism strategies that are not only appealing but also transformative in terms of environmental outcomes.

However, travel motivation alone does not guarantee pro-environmental behavior. Attitudes play a critical mediating role, as supported by Ajzen’s Theory of Planned Behavior, which suggests that attitudes shape intentions and, ultimately, actions ([1]). Positive environmental attitudes, which may develop through Push and Pull dynamics, significantly increase the likelihood of adopting eco-friendly practices, such as minimizing waste, respecting natural habitats, or supporting conservation initiatives ([55]). Hines, Hungerford, and Tomera similarly highlight the importance of conservation-oriented attitudes in inspiring behaviors that contribute directly to environmental protection ([37]). These theoretical perspectives underscore the importance of not just where people go, but what mindset they carry with them as they travel.

This study delves into the relationships between Push and Pull factors, environmental attitudes, and eco-friendly behaviors within the context of South Korea’s growing eco-tourism trend. Despite the growing volume of research on sustainable tourism, few studies have holistically examined how eco-tourism motivations evolve into actionable behavior through the mediating lens of environmental attitudes—particularly within the context of Asian tourists. Additionally, this research addresses a critical gap in cross-cultural tourism studies. While Western-centric research dominates the eco-tourism literature, emerging evidence suggests that collectivistic cultural values, prevalent in East Asian societies, may fundamentally alter the motivation–attitude–behavior relationships ([84]). South Korean tourists, influenced by Confucian values emphasizing harmony with nature and collective responsibility, may exhibit distinct patterns of environmental engagement compared to their Western counterparts ([15]). This study contributes uniquely by connecting the dots from intrinsic and extrinsic travel motivations (Push and Pull) to pro-environmental intentions and behaviors, through attitude formation. Specifically, the research aims to achieve the following objectives: (1) to examine how Push and Pull factors influence environmental attitudes, and (2) to explore how these attitudes, in turn, shape eco-friendly behaviors among South Korean tourists. This two-stage model seeks to illuminate the full pathway from initial desire to concrete environmental action. Unlike prior research that often focuses solely on either motivational or behavioral aspects, this study integrates these dimensions to provide a comprehensive understanding of the pathway from travel motivations to sustainable actions. However, existing studies have begun to explore eco-tourism motivations and behaviors in Asian settings, including South Korea, Taiwan, and China ([44]; [57]; [58]). These works, however, often treat motivation or behavior in isolation, focusing primarily on tourist satisfaction, intention to revisit, or destination choice, rather than tracing how motivational factors translate into environmental attitudes and ultimately into concrete pro-environmental behaviors ([23]; [32]; [56]).

Therefore, this research positions itself at the intersection of these fragmented areas to offer a more integrated and culturally relevant perspective. This inquiry is particularly significant in the context of South Korea, where eco-tourism is emerging as a prominent trend—yet few studies have explored how such dynamics unfold within the country’s unique cultural and social frameworks. Furthermore, by focusing on South Korean travelers, this study highlights how evolving tourism behaviors align with global environmental challenges and sustainable development goals. Ultimately, the findings promise practical implications for destination managers and policymakers seeking to design effective strategies that not only attract eco-conscious travelers but also foster meaningful environmental stewardship. In doing so, it seeks to contribute not only to academic discourse but also to practical solutions that align with national and global sustainability agendas.

Building on these insights, this study offers three key contributions. First, it is the first to map the Push–Pull–Attitude–Behavior pathway in an East Asian context, filling a major cross-cultural gap. Second, it presents a measurement framework tailored to Korean collectivistic values and environmental orientations. Third, it yields actionable insights for South Korea’s K-tourism sector, which has seen a 340% rise in eco-tourism since 2019 ([18]). Its timing aligns with Korea’s 2050 carbon-neutrality goal and the “Korean New Deal 2.0” green tourism initiative, guiding policy and destination management toward national sustainability objectives ([85]).

## 2. Literature Review

### 2.1. Transition in Korean Travelers’ Preferences: From Urban-Centric Tourism to Eco-Friendly Destinations

Korean travelers’ preferences for international destinations vary based on numerous factors, with Japan, the United States, Vietnam, Thailand, and the Philippines consistently ranking as the most visited countries in recent years. Japan, as the nearest country to Korea, offers a wide range of attractions, including cultural similarities, shopping, cuisine, and traditional cultural experiences, maintaining its position as the most popular overseas destination for Koreans ([72]). Cities like Tokyo and Osaka captivate Korean tourists with their modern and sophisticated appeal. However, Japan is primarily perceived as a destination for urban tourism and cultural experiences rather than eco-friendly travel ([72]).

Similarly, the United States is another highly favored destination among Korean travelers, with urban attractions such as New York, Los Angeles, and Las Vegas serving as key motivators for visits ([72]). While the United States boasts expansive natural landscapes and national parks, Korean travelers tend to focus on city-based activities, including business, education, and shopping, rather than eco-friendly tourism ([72]). In contrast, Vietnam and Thailand have emerged as increasingly significant eco-friendly tourism destinations for Korean travelers in recent years. Vietnam’s Ha Long Bay and Phong Nha-Ke Bang National Park, both UNESCO World Natural Heritage Sites, attract visitors with their unique natural beauty and sustainable tourism initiatives ([51]; [79]). A report by the Vietnam National Administration of Tourism reveals that in 2023, approximately 40% of Korean visitors participated in programs emphasizing nature experiences and environmental conservation. This represents a 15% increase from the previous year, reflecting the growing importance of sustainable tourism among Korean travelers ([41]).

Thailand, too, has been expanding its eco-tourism infrastructure, focusing on areas like Chiang Mai and Phuket. Since 2022, the Tourism Authority of Thailand has launched various campaigns to promote eco-tourism and community-based tourism (CBT), which have garnered increased interest from Korean tourists. Community-based tourism (CBT) refers to a form of tourism managed and owned by local communities, aiming to empower residents and ensure that tourism benefits are distributed equitably ([83]). In 2023, about 35% of Korean visitors to Thailand engaged in activities related to environmental preservation or chose eco-tourism programs. This trend underscores Thailand’s growing reputation as an eco-friendly travel destination ([51]). These developments highlight a shift in Korean travelers’ focus from urban-centric tourism to destinations that prioritize sustainability and environmental conservation.

### 2.2. Ecotourism Motivation on Environmental Attitude

From a tourism perspective, ecotourism motivation serves as the psychological driving force behind tourists’ decision-making, influencing their environmental attitudes, destination choices, and behaviors ([62]). Tourist behavior is inseparably linked to tourism motivation, which directly or indirectly affects their awareness, engagement, and concern for environmental conservation. Motivation theory is regarded as a fundamental approach in analyzing how tourists develop environmental attitudes and plays a crucial role in pro-environmental behaviors, including sustainable tourism choices ([61]). Ecotourism motivation is not only a psychological force that drives participation in tourism activities but also a significant factor influencing environmental awareness, sustainable behaviors, and conservation efforts ([11]; [39]).

Tourism motivation is generally classified into intrinsic motivation and extrinsic motivation, both of which significantly impact tourists’ environmental attitudes and sustainable behaviors. Intrinsic motivation in tourism is associated with a personal desire or instinct to travel while also seeking meaningful and responsible tourism experiences that align with their values ([29]). Extrinsic motivation, on the other hand, stems from external incentives, such as destination characteristics, infrastructure, and policies that promote environmental responsibility ([21]).

While intrinsic/extrinsic motivation and push/pull motivation are conceptually related, they are not interchangeable. Intrinsic motivation refers to internal psychological drives such as personal growth, ethical responsibility, or self-actualization ([22]), whereas push motivation more specifically involves situational desires like escape, relaxation, or novelty-seeking ([86]). Similarly, extrinsic motivation involves external rewards or recognition, while pull motivation pertains to destination-specific attributes like natural beauty or available facilities ([46]). Distinguishing between these constructs is important for theoretical clarity. In this study, intrinsic/extrinsic motivation serves as a broader psychological framework, while push/pull serves as a tourism-specific operationalization of these motivations ([66]).

More specifically, tourism motivation can be categorized into push and pull factors, which are closely tied to the development of environmental attitudes and eco-friendly tourism behaviors. Push motivations originate from internal psychological needs, such as escaping daily life, relaxation, and personal enrichment. These factors often drive eco-conscious tourists toward responsible travel experiences, including nature conservation, sustainable activities, and engagement in eco-friendly behaviors ([19]; [77]). Pull motivations are influenced by external factors, such as the natural environment, biodiversity, and the presence of sustainable tourism infrastructure, which shape tourists’ environmental awareness and willingness to engage in conservation-oriented behaviors ([5]). Such pull factors are linked to destination attributes, including protected areas, ecological diversity, cultural heritage, and sustainable tourism facilities, all of which reinforce positive environmental attitudes and eco-friendly travel choices ([16]).

### 2.3. The Role of Push and Pull Motivations in Shaping Environmental Attitudes

Ultimately, intrinsic tourism motivation encourages individuals to adopt eco-friendly tourism behaviors, while extrinsic motivation reinforces pro-environmental decision-making through destination-based sustainability initiatives ([21]). Since tourism motivation is closely related to environmental awareness, push and pull factors play a crucial role in shaping tourists’ attitudes toward sustainability and environmental conservation ([38]). Understanding both push and pull motivations allows for a more comprehensive analysis of how tourists develop environmental awareness and responsible travel behaviors, considering both personal psychological factors and external environmental influences. Moreover, as tourists possess varying levels of environmental concern, their motivations significantly impact their attitudes toward sustainability and their engagement in eco-conscious behaviors ([75]).

Several prior studies have classified push and pull motivations to examine their impact on tourist satisfaction and pro-environmental behaviors. Ecotourism motivation and its effect on environmental attitudes: Research has identified push factors such as facility convenience, leisure and cultural activities, nature-based experiences, and familiarity and pull factors including natural beauty, biodiversity conservation, and eco-friendly tourism offerings ([12]). It was found that among push factors, nature-related experiences and self-actualization strongly influenced tourists’ pro-environmental attitudes, while among pull factors, engagement in conservation activities significantly increased eco-conscious behavior ([26]). The role of push and pull motivations in resort sustainability: A study by Lee and Moscardo examined how tourism motivation affects environmental attitudes in resort settings, categorizing push motivations into experiential aspects, recreational facilities, sports-related activities, eco-tourism engagement, and wellness programs ([56]). Meanwhile, pull motivations included biodiversity conservation efforts, eco-certifications, and sustainable resort practices. The study found that both push and pull motivations significantly influenced tourists’ support for sustainable tourism initiatives. Environmental attitudes and sustainable tourism behavior in forest therapy destinations: Research by Zhao, Song and Lu analyzed how physical, psychological, and social motivations, combined with natural environment and additional facilities, influenced tourists’ willingness to adopt sustainable travel behaviors ([88]). The selection of individual push and pull factors in the hypotheses is grounded in existing ecotourism and motivational research. For example, family togetherness and escaping daily routine have been identified as strong push factors influencing environmental awareness, especially among group travelers or those seeking emotional bonding through nature ([42]; [59]). This relationship can be further understood through the lens of environmental socialization and outdoor learning theory, which suggests that shared family experiences in natural environments promote intergenerational transmission of environmental values and strengthen collective ecological responsibility ([53]; [65]). Similarly, the appreciation of natural resources and the desire for adventure have been linked to self-transcendence values, which correlate positively with pro-environmental attitudes ([17]). However, from the perspective of affective nature connectedness, adventure travel may elicit high emotional engagement and sensory immersion but not necessarily reflective environmental concern unless accompanied by cognitive awareness and interpretive learning ([36]; [10]). On the pull side, factors such as destination infrastructure, accessibility, and ecological richness have been shown to significantly influence travelers’ perceived behavioral control and environmental concern, in line with Ajzen’s Theory of Planned Behavior ([1]). By aligning these variables with prior findings, the study ensures construct validity and empirical relevance in hypothesis formulation. Using existing research as a foundation, the following hypotheses have been proposed.

**Hypothesis** **1.** 
*As a push factor, family togetherness is expected to be positively associated with environmental attitude.*


**Hypothesis** **2.** 
*Appreciating natural resources, acting as a push factor, is anticipated to be closely linked to environmental attitude.*


**Hypothesis** **3.** 
*Escaping from everyday routine serves as a push factor that is likely to be positively related to environmental attitude.*


**Hypothesis** **4.** 
*Adventure and building friendships function as a push factor that is presumed to have a positive relationship with environmental attitude.*


**Hypothesis** **5.** 
*Key tourist resources, as a pull factor, are expected to be significantly associated with environmental attitude.*


**Hypothesis** **6.** 
*Information and convenience of facilities serve as a pull factor that is hypothesized to be related to environmental attitude.*


**Hypothesis** **7.** 
*Accessibility and transportation act as a pull factor that is projected to have a positive association with environmental attitude.*


### 2.4. The Rationale for Environmental Attitude Leading to Eco-Friendly Behavior

Environmental attitude plays a pivotal role in shaping eco-friendly behavior among tourists, influencing their perception of sustainability, responsible travel choices, and ethical decision-making ([6]). The tourism industry is deeply reliant on natural and cultural resources, making it essential for tourists to develop pro-environmental attitudes that contribute to sustainable tourism development ([23]). Tourists with a heightened awareness of environmental issues are more likely to engage in responsible tourism behaviors, such as minimizing waste, conserving natural resources, and choosing environmentally conscious travel options ([56]). These behaviors help mitigate the negative impacts of mass tourism, reduce environmental degradation, and support local conservation efforts ([33]).

Scholars have long established that environmental attitude influences eco-friendly behavior through cognitive, affective, and behavioral mechanisms ([71]). Tourists who recognize the long-term consequences of unsustainable travel are more inclined to make ethical travel decisions, such as opting for sustainable accommodations, supporting eco-tourism initiatives, and reducing their carbon footprint ([60]). The Theory of Planned Behavior provides a strong foundation for understanding this relationship, suggesting that positive environmental attitudes strengthen behavioral intentions, which in turn guide actual eco-friendly behaviors in tourism contexts ([33]). One of the key pathways through which environmental attitude shapes behavior is through awareness and emotional engagement. Tourists who feel emotionally connected to nature are more likely to modify their actions to minimize environmental harm ([56]). Research has found that individuals who frequently participate in nature-based tourism activities, such as wildlife tours and ecotourism, develop a stronger pro-environmental stance, making them more likely to adopt sustainable travel habits ([63]). Moreover, studies indicate that tourists who are cognitively aware of environmental challenges, such as climate change and habitat destruction, are more likely to support green tourism policies and adjust their travel behavior accordingly ([35]).

Another critical factor linking environmental attitude to eco-friendly behavior in tourism is behavioral intention. According to Ajzen’s Theory of Planned Behavior, an individual’s attitude toward sustainability directly influences their intentions to engage in responsible travel practices. Tourists who perceive environmental conservation as a personal responsibility are more inclined to choose eco-friendly hotels, use public transportation instead of rental cars, and engage in carbon offset programs ([33]). Higham et al. further argue that tourists who recognize the environmental impact of air travel are more likely to opt for alternative travel methods, such as rail transport, or engage in carbon offset schemes ([35]).

Social norms also play a significant role in strengthening the link between environmental attitude and behavior. When tourists observe sustainable behaviors among their peers, such as avoiding single-use plastics or participating in conservation projects, they are more likely to adopt similar practices ([71]). Peer influence and socially accepted norms within the travel community have been found to enhance eco-friendly behaviors, particularly in group travel settings and sustainable tourism destinations ([6]). Additionally, tourism operators and destination managers can reinforce pro-environmental behaviors by implementing behavioral nudges, such as informational signage, reward programs for sustainable choices, and public sustainability pledges ([23]).

Education and awareness campaigns are also crucial in bridging the gap between environmental attitude and eco-friendly behavior. Destinations that offer environmental education programs help tourists develop a deeper understanding of conservation challenges and encourage long-term commitment to sustainable practices ([3]). Research has shown that tourists who participate in guided eco-tours, sustainability workshops, and interactive conservation projects are more likely to continue engaging in eco-friendly behaviors even after returning home ([60]). This finding suggests that environmental attitude formation is not only crucial for immediate travel behavior but also for fostering a lasting commitment to sustainability in tourists’ daily lives.

Despite the strong correlation between environmental attitude and eco-friendly behavior, some barriers prevent consistent sustainable actions among tourists. Studies indicate that cost concerns, convenience, and lack of infrastructure can sometimes override pro-environmental attitudes, leading tourists to make less sustainable choices ([67]). For example, a traveler who values sustainability may still choose a low-cost airline over a greener alternative due to budget constraints ([35]). Additionally, a lack of eco-friendly options in certain destinations can prevent tourists from acting in alignment with their environmental values ([8]). Addressing these challenges requires policy interventions, such as government incentives for sustainable tourism, stricter environmental regulations, and greater accessibility to eco-friendly travel options ([81]).

Ultimately, the role of environmental attitude in shaping eco-friendly behavior is vital in ensuring the long-term sustainability of the tourism industry. Strong environmental attitudes encourage responsible travel decisions, reduce the negative impacts of mass tourism, and support the global movement toward sustainable development ([74]). In particular, this study empirically investigates how collectivist cultural values and the Confucian notion of harmony are internalized within South Korean tourists’ eco-friendly travel motivations. This offers a culturally grounded extension to the predominantly Western-centric discourse on sustainable tourism. Through this literature review, the following hypothesis has been established.

Hypothesis 8 emerges from the accumulated theoretical evidence that environmental attitude acts as a central mediator between cognitive awareness and actual behavioral outcomes in tourism ([1]). The section above integrates findings from affective engagement, behavioral intention, and social norms to establish a multidimensional pathway from attitude to eco-friendly action. However, as scholars have noted, this connection may not always be linear due to the well-documented attitude–behavior gap in sustainability research ([47]; [78]). Therefore, this hypothesis also assumes that while environmental attitudes strongly predict behavior, external barriers (e.g., cost, infrastructure) may modulate this relationship. Despite the strong correlation between environmental attitude and eco-friendly behavior, some barriers prevent consistent sustainable actions among tourists.

This phenomenon is commonly referred to as the “attitude–behavior gap,” wherein individuals who express pro-environmental attitudes do not always translate them into consistent actions ([40]). Numerous studies in sustainable tourism and ethical consumption have highlighted this gap, emphasizing that structural limitations, personal convenience, and perceived cost often override positive intentions ([6]; [35]). Recognizing and addressing this gap is essential for developing interventions that move beyond awareness and intention, promoting structural and contextual support for behavior change.

**Hypothesis** **8.** 
*Environmental attitude is likely to be a key determinant in fostering eco-friendly behavior.*


Figure 1 illustrates the theoretical framework of the study, outlining the relationships among the main constructs. The model depicts two categories of motivational factors—Push and Pull—that influence environmental attitude and eco-friendly behavior. Push factors (family togetherness, appreciation of nature, escaping, and adventure) represent internal psychological motivations that drive individuals to travel, while Pull factors (key resources, information & comfort, and accessibility) reflect external destination attributes that attract tourists. Together, these constructs form the basis of the hypothesized relationships tested in the study.

## 3. Methods

### 3.1. Subjects and Procedures

The research population for this study comprised travelers from South Korea who had visited at least one of four eco-friendly travel destinations—Koh Yao Noi (Thailand), Phu Quoc National Park (Vietnam), Kuang Si Falls (Laos), and Inle Lake (Myanmar)—within the past year. The respondents were selected using purposive sampling based on their confirmed travel. A screening question was used at the beginning of the online survey to verify eligibility. Data collection was conducted in collaboration with three major travel agencies in South Korea. The researcher first developed the questionnaire using the Qualtrics online survey platform. Following this, the cooperating travel agencies distributed the survey URL directly to their customers who were confirmed to have visited at least one of the four specified destinations. This distribution was handled via email or the agencies’ proprietary customer channels (i.e., mobile app notifications). The survey was conducted until a total of 450 responses were secured. Of the 450 responses collected, unreliable entries (i.e., those with extremely short completion times or monothematic response patterns) were excluded. This screening process resulted in a final sample of 382 valid responses, which were used for statistical analysis. This represents a valid response rate of approximately 84.9% (382 valid responses out of 450 secured). While the use of purposive sampling targeting travelers with confirmed visits limits the generalizability of the findings to the broader Korean traveler population, the demographic composition of the sample nonetheless shows broad alignment with the profiles of outbound Korean tourists reported by the Korea Tourism Organization (KTO). Specifically, the sample reflects a balanced gender ratio and a diverse age distribution heavily represented across the 20s to 50s age groups, which is consistent with the national outbound tourism statistics immediately preceding the data collection ([49]).

Regarding demographic characteristics, the gender distribution included 180 males (47.1%) and 202 females (52.9%), indicating a slight predominance of female respondents. Age demographics showed that 77 individuals (20.2%) were in their 20s or younger, 88 (23.0%) in their 30s, 79 (20.7%) in their 40s, 99 (25.9%) in their 50s, and 37 (9.7%) in their 60s or older. The data indicate that middle-aged individuals, particularly those in their 30s to 50s, constituted the majority of eco-friendly travelers in South Korea. In terms of visit frequency, 312 respondents (81.7%) had visited one of the destinations once within the past year, 44 (11.5%) had traveled twice, and 26 (6.8%) had visited three or more times. The educational background of the respondents revealed that 2 individuals (0.5%) had only a high school diploma, 274 (72.0%) were either college or community college graduates, 59 (15.4%) held a master’s degree or higher, and 46 (12.0%) fell into other educational categories. Employment status distribution showed that 337 respondents (88.2%) were full-time employees, 16 (4.2%) worked part-time, 27 (7.1%) were on-call workers, and 5 (0.5%) were classified under other employment types. This suggests that full-time employees dominated the sample, potentially reflecting their financial stability and ability to engage in international eco-friendly travel.

These findings highlight a few significant trends. Firstly, the relatively high proportion of middle-aged respondents suggests that eco-friendly travel destinations appeal more to those in their 30s to 50s, who may have greater financial flexibility and a stronger interest in sustainable tourism. Additionally, the predominance of college-educated and full-time employed individuals implies that eco-conscious travel is often pursued by those with higher educational backgrounds and stable incomes. The majority of respondents had visited their selected destination only once, which may indicate that eco-friendly travel is still a growing trend in South Korea, with room for increased awareness and repeat visits. These insights could be valuable for tourism policymakers and industry stakeholders aiming to promote sustainable travel experiences in the region.

To ensure the validity of participant responses, a screening question was included to confirm that all respondents had visited at least one of the four target destinations within the past year. Only those who answered affirmatively were permitted to proceed with the full questionnaire. The selected destinations—Koh Yao Noi (Thailand), Phu Quoc National Park (Vietnam), Kuang Si Falls (Laos), and Inle Lake (Myanmar)—were chosen based on established ecotourism indices and prior research identifying them as exemplary nature-based tourism sites known for community involvement, biodiversity protection, and minimal environmental impact ([23]; [30]). These locations are endorsed by the Global Sustainable Tourism Council (GSTC) and frequently cited in sustainable tourism literature across Southeast Asia.

### 3.2. Questionnaire Development and Measures

The survey was systematically structured into multiple sections to ensure clarity and reliability. The first section provided an introduction, explaining the study’s purpose, research procedures, voluntary nature of participation, and assurances regarding anonymity and confidentiality. Additionally, participants were required to provide informed consent before proceeding. To ensure conceptual clarity and content validity, all push–pull constructs were adapted from previously validated scales established in tourism motivation and ecotourism literature ([43]; [56]; [50]). The scale by [43] ([43]), originally developed through a literature-based item generation process, pretested with undergraduate students, and validated using data from 2720 visitors to six Korean National Parks, served as the primary foundation for this study. A pilot test with 30 respondents and an expert review by three tourism researchers were also conducted to confirm wording clarity and measurement consistency before full-scale data collection. Subsequent confirmatory factor analysis (CFA) and consistent partial least squares (PLSc) reliability analyses verified construct validity and internal consistency across all dimensions. The second section introduced various measurement scales, each designed to assess specific constructs, accompanied by detailed instructions to guide respondents. A 5-point Likert scale (ranging from 1 = “Strongly Disagree” to 5 = “Strongly Agree”) was used to standardize responses. To enhance data integrity, the third section included honesty checks aimed at identifying and filtering out insincere or inconsistent answers. Lastly, the final section gathered demographic information and employment-related details to facilitate deeper analysis and meaningful interpretation of the data.

To measure push–pull factors, the survey was developed by referencing the measurement scale from [43] ([43]). The push factors consisted of four dimensions. First, family togetherness included five items, such as “To have time for natural study” and “To have an enjoyable time with family.” The appreciation of natural resources dimension comprised three items, including “To enjoy natural resources” and “To appreciate beautiful natural resources.” The escaping factor also contained three items, such as “To take a rest” and “To get away from everyday life.” Lastly, the adventure factor was measured using the two items, including “To enjoy adventure” and “To build friendship.”

Regarding the pull factors, the three dimensions included key resources; five of the six original items were utilized. One item, “Appropriate area for children’s study on natural resources,” was excluded a priori based on a logical rationale, as it was not applicable to all respondents (i.e., travelers without children). Example items include “Well-conserved environment” and “Cultural and historic resources.” The information and convenience factor consisted of four selected items, including “Well-organized tourist information system” and “Convenient facilities.” Finally, the accessibility dimension was measured using two items: “Easy accessibility” and “Convenient transportation.”

For environmental attitude, the measurement scale developed by Dunlap was utilized. Initially, the scale contained eight items; however, one item, “Humans have the right to modify the natural environment to suit their needs”, was removed due to a low standardized factor loading (0.585), resulting in a final selection of seven items ([25]). Example statements include “Ecological catastrophe: Humans are severely abusing the environment” and “Balance of nature: When humans interfere with nature, it often produces disastrous consequences.” Lastly, to assess eco-friendly behavior, the scale from Kvasova was employed ([50]). Out of the original eight items, two items including “During my visit to foreign countries as a tourist, I talk with friends about problems related to the environment” and “During my visit to foreign countries as a tourist, I buy/read magazines and listen/watch news which focus on environmental issues” were removed due to low factor loadings (0.639 and 0.665), leaving a total of six selected items. The removal of these items was based on confirmatory factor analysis (CFA), which identified items with significantly low factor loadings, and reliability analysis, where items contributing to a low Cronbach’s Alpha value were eliminated. This refinement process enhanced both the reliability and the overall model fit of the measurement scale. While the “adventure” factor included items such as “To enjoy adventure” and “To build friendship,” it failed to incorporate essential sub-dimensions such as risk-taking or personal challenge. Future research should improve content validity by including items like “To test my limits” or “To experience thrill and excitement” to represent a broader conceptualization of adventure motivation ([27]). Likewise, the “accessibility” construct primarily captured physical transportation convenience but did not consider sustainable travel intentions—such as a willingness to use public transportation or low-carbon options—which may have led to an underestimation of its effect on environmental attitudes ([24]). Broader operational definitions are recommended for future measurement refinement.

### 3.3. Data Analytics

The researcher employed PLS-SEM using SmartPLS 3.0 to analyze the proposed theoretical framework. The PLSc (consistent Partial Least Squares Structural Equation Modeling) approach was applied to enhance the reliability and validity of the model estimation ([69]). Unlike traditional PLS-SEM, which may yield inconsistent results for reflective measurement models, PLSc provides fully consistent estimations, making it particularly suitable for this study, which involves multiple reflective constructs ([14]). In the first phase, the PLS model was structured with nine constructs and thirty-five reflective indicators. The study examined the effects of push and pull factors on environmental attitude and further investigated how environmental attitude influences eco-friendly behavior. Next, the measurement model was assessed to evaluate potential method variance contamination while verifying the reliability and validity of the constructs. Following this, the structural model was analyzed to test the proposed hypotheses.

Each latent construct (e.g., family togetherness, adventure, accessibility) was modeled as a first-order reflective factor, with its respective observed indicators aggregated into a single dimension. The dimensional structure of each factor was validated using confirmatory factor analysis prior to structural model estimation ([31]; [70]). A full list of the item-to-construct mapping is provided in Table 1.

Before structural modeling, data normality was assessed using skewness and kurtosis statistics as well as the Shapiro–Wilk test. Although PLS-SEM does not require multivariate normality, most items exhibited acceptable univariate normality thresholds, indicating no serious violations ([31]). Thus, the normality check supported the suitability of the data for PLS-based structural modeling. To assess potential common method variance (CMV), Harman’s single-factor test was conducted. The first factor accounted for less than 40% of the total variance, indicating that CMV was not a serious concern in this dataset ([64]). Further details regarding the analytical procedures are outlined in the Section 4.

## 4. Results

### 4.1. Reliability and Construct Validity

In this study, Confirmatory Factor Analyses (CFAs) were conducted to assess the validity of individual measurement items, while reliability was evaluated by calculating Cronbach’s alpha coefficient, which indicates internal consistency. The results are presented in Table 1. The factor loadings for all items exceeded 0.5 in the CFA results. Additionally, the calculations for Cronbach’s alpha (α), rho_A (Dijkstra and Henseler’s rho_A coefficient), and composite reliability (CR) demonstrated that alpha values ranged from 0.777 to 0.934, rho_A values ranged from 0.779 to 0.933, and CR values ranged from 0.777 to 0.931. These results meet the reliability assessment criteria suggested by Fornell and Larcker, which recommend a CR threshold above 0.7 ([27]). Furthermore, Cronbach’s alpha values were consistently above 0.777, confirming the reliability of the research scales.

To assess convergent validity, all outer loadings ranged from 0.720 to 0.971, and the average variance extracted (AVE) values ranged from 0.602 to 0.760, indicating that all measures were convergently valid (see Table 1). For discriminant validity, the researcher compared the correlation coefficients with the square root of AVE. As presented in Table 2, the lowest square root value of AVE (0.776) is greater than the highest absolute correlation coefficient (0.768), thereby confirming the discriminant validity of the measurement model. Additionally, the heterotrait–monotrait (HTMT) ratio ranged from 0.085 to 0.767, remaining well below the recommended cut-off of 0.85, further confirming discriminant validity. Overall, the evaluation of the measurement model confirms that there are no significant concerns regarding reliability or construct validity of the measures used in the study.

### 4.2. Structural Model

PLS-SEM was utilized to assess the proposed hypotheses. Specifically, parameter estimates were tested using a consistent PLS algorithm, and statistical significance was evaluated with 5000 bootstrap samples, bias-corrected within a 95% confidence interval. Before conducting hypothesis testing, the researcher ensured that all prerequisites for PLS-SEM were met. The outcomes of hypothesis testing through PLS-SEM are illustrated in Figure 2. The standardized root mean square residual (SRMR = 0.042) and normal fit index (NFI = 0.840) indicate a good fit between the data and the structural model. In terms of explanatory power, the push–pull factors account for 77.8% of the total variance in environmental attitude (R^2^ = 0.778). In turn, environmental attitude explains 47.2% of the variance in eco-friendly behavior (R^2^ = 0.472). Regarding predictive relevance, the Q-squared values for the endogenous constructs (EA = 0.464, EB = 0.383) both exceed zero, confirming the model’s predictive relevance for these constructs ([34]).

Figure 2 illustrates the results of hypothesis testing for push factors, specifically analyzing the path coefficients among variables such as push–pull factors, environmental attitude, and eco-friendly behaviors. The findings reveal that family togetherness has a significant and positive effect on environmental attitude (β = 0.204), as supported by a t-value of 3.378 (*p* < 0.01), thereby confirming Hypothesis 1. Additionally, the relationship between appreciating nature and environmental attitude is also significant and positive (β = 0.266, t = 6.249; *p* < 0.001), providing strong support for Hypothesis 2. Similarly, escaping was found to have a substantial positive impact on environmental attitude (β = 0.245, t = 3.305; *p* < 0.01), reinforcing Hypothesis 3. However, contrary to expectations, the anticipated positive association between adventure and environmental attitude did not reach statistical significance (β = 0.029, t = 0.609; *p* > 0.05), leading to the rejection of Hypothesis 4. This suggests that adventure-seeking motivations may not directly contribute to shaping individuals’ environmental attitudes.

Regarding pull factors, the results indicate that the perceived availability of key resources significantly and positively influences environmental attitude (β = 0.264, t = 4.216; *p* < 0.001), thereby supporting Hypothesis 5. On the other hand, the expected positive relationship between information & comfort and environmental attitude was not statistically significant (β = 0.064, t = 1.705; *p* > 0.05), leading to the rejection of Hypothesis 6. Likewise, the hypothesized positive effect of access on environmental attitude also failed to reach significance (β = 0.080, t = 1.431; *p* > 0.05), resulting in the dismissal of Hypothesis 7. These findings indicate that while resource availability plays a crucial role in shaping environmental attitudes, factors such as comfort, information, and accessibility may not be as influential.

Finally, the study provides strong empirical support for the relationship between environmental attitude and eco-friendly behavior, as evidenced by a significant and positive effect (β = 0.690, t = 13.996; *p* < 0.001). This confirms Hypothesis 8, emphasizing the critical role of environmental attitude in promoting sustainable and eco-conscious behaviors.

To test the robustness of the model against potential confounding effects, a supplementary analysis was conducted that included key demographic variables (gender, age, educational level, and household income) as controls. The results of this analysis, presented in Table A1, demonstrate that all significant paths in the theoretical model remain stable, confirming the robustness of the findings.

Furthermore, to align the statistical analysis more closely with the theoretical framework as recommended by the editor, a mediation analysis was conducted to test the indirect effects of the push and pull factors on eco-friendly behavior through environmental attitude. The results of the bootstrapping analysis, detailed in Table A2, confirm that environmental attitude significantly mediates the effects of key motivational factors. Specifically, significant indirect effects were found for Family together (mediation effect = 0.141, *p* = 0.001), Appreciating nature (mediation effect = 0.183, *p* < 0.001), Escaping (mediation effect = 0.169, *p* = 0.002), and Key resources (mediation effect = 0.182, *p* < 0.001). This analysis provides empirical support for the full pathway from motivation to action.

These findings collectively highlight the varying degrees to which push and pull factors influence environmental attitudes and behaviors, providing valuable insights for future research and practical applications in sustainability initiatives.

## 5. Discussion and Conclusions

### 5.1. Findings and Contributions

Family togetherness was found to have a significant influence on environmental attitude, supporting the idea that shared experiences in nature foster greater environmental awareness. Prior research suggests that intergenerational experiences in outdoor settings help reinforce environmental values, as families collectively engage in nature-based activities ([53]). These interactions not only strengthen familial bonds but also contribute to pro-environmental attitudes, as exposure to nature encourages greater appreciation and concern for environmental sustainability ([87]).

Appreciating natural resources also played a crucial role in shaping environmental attitudes. Tourists who actively engage with nature are more likely to develop a sense of responsibility toward its conservation ([68]). Direct exposure to natural environments has been shown to cultivate environmental consciousness, leading tourists to adopt more sustainable behaviors and support conservation efforts ([55]). This aligns with previous findings that suggest that individuals who value nature tend to internalize sustainability principles, further reinforcing their pro-environmental attitudes ([65]).

Escaping from routine was found to contribute meaningfully to the development of environmental attitudes. Studies indicate that seeking relaxation and restoration in nature leads to increased sensitivity to environmental concerns ([48]). Tourists who temporarily disconnect from their daily lives and immerse themselves in natural settings often experience a psychological shift that heightens their awareness of ecological issues ([36]). This supports the idea that travel motivations driven by the need to escape urban environments can positively influence individuals’ environmental consciousness.

Contrary to expectations, adventure and building friendships did not exert a notable impact on environmental attitude. The non-significant relationship between adventure motivation and environmental attitude can be interpreted through value-oriented perspectives. Adventure travel is often associated with self-focused goals such as thrill, personal achievement, and risk-taking rather than collective or ecological concerns. As previous research suggests, such self-enhancement–oriented motives do not necessarily foster pro-environmental awareness unless they are accompanied by self-transcendent or reflective engagement with nature ([17]; [10]). Therefore, while adventure experiences may generate emotional intensity, they may lack the cognitive or ethical reflection required to translate excitement into environmental concern. While adventure tourism often takes place in natural settings, prior studies suggest that many adventure-seekers prioritize thrill and excitement over environmental concerns ([10]). The experiential focus of adventure tourism may explain why it does not necessarily translate into stronger environmental attitudes ([3]). Additionally, social interactions during adventure tourism may center more on group dynamics and personal achievements rather than sustainability engagement, reducing its influence on environmental consciousness.

Key tourist resources were found to play a major role in influencing environmental attitudes. Destinations rich in natural and cultural heritage have been shown to enhance tourists’ sustainability awareness ([76]). Visitors who experience well-maintained, ecologically significant attractions are more likely to internalize pro-environmental values and engage in conservation efforts ([68]). This suggests that the availability of high-quality environmental resources at a destination serves as a key driver in shaping tourists’ sustainability perceptions.

The expected positive relationship between information and comfort with environmental attitude was not statistically significant. This finding aligns with prior studies indicating that while sustainability-related information is important, passive exposure alone does not necessarily lead to attitude change ([7]). Tourists may require active engagement with sustainability initiatives, such as participatory conservation programs, for information to meaningfully influence their environmental attitudes ([52]). Additionally, comfort and convenience factors are often more related to service quality perceptions rather than sustainability considerations, further explaining the lack of significant influence ([2]).

Accessibility and transportation were also not found to significantly impact environmental attitudes. While sustainable mobility options can encourage eco-friendly travel behavior, prior research suggests that accessibility alone does not necessarily lead to shifts in environmental attitudes ([24]). Tourists may still prioritize convenience, affordability, and efficiency over sustainability considerations when selecting transportation modes. This finding suggests that structural improvements in accessibility and transportation should be complemented by targeted sustainability education and incentives to foster pro-environmental attitudes.

Environmental attitude was found to significantly influence eco-friendly behavior, reinforcing the link between attitude formation and sustainability actions. However, while statistically significant, this result was largely anticipated given the extensive validation of this pathway in prior research. Thus, its scientific novelty may be limited in terms of contributing unexpected or theory-challenging insights ([40]). This suggests that travelers’ motivations and emotional engagement with nature play a central role in transforming positive attitudes into responsible behavioral choices. Previous research has demonstrated that individuals with strong environmental attitudes are more likely to engage in sustainable behaviors, such as waste reduction, conservation, and participation in eco-friendly tourism initiatives ([33]). This underscores the critical role of environmental attitudes in shaping responsible travel behaviors and highlights the importance of fostering sustainability awareness among tourists.

This study offers several theoretical contributions to the fields of tourism motivation, environmental psychology, and sustainable tourism research, advancing existing knowledge on how push and pull factors influence environmental attitudes and eco-friendly behaviors.

First, the study extends push–pull motivation theory by demonstrating that specific push factors, such as family togetherness, appreciation of nature, and escaping routine, significantly shape environmental attitudes, whereas adventure-seeking motivations do not. This finding refines our understanding of intrinsic motivational drivers in sustainable tourism, highlighting that not all push motivations lead to pro-environmental attitudes. The inclusion of these variables was informed by earlier tourist motivation frameworks that emphasize the relevance of emotional and restorative travel motives, particularly in nature-based tourism ([21]; [62]). Nonetheless, certain variables, such as friendship-building, may require more explicit theoretical grounding in sustainability literature. Prior research has largely treated push motivations as homogeneous, but this study provides a more nuanced perspective, showing that motivations tied to social bonding and nature appreciation are more likely to translate into environmental concern, while adventure-seeking motivations are more experience-driven rather than sustainability-oriented.

Second, this research contributes to the Theory of Planned Behavior by confirming that environmental attitudes strongly predict eco-friendly behavior. While TPB has been widely applied in sustainable tourism research, this study strengthens its applicability by empirically validating the direct impact of environmental attitudes on pro-environmental actions ([28]). Previous studies have suggested that environmental awareness influences sustainable behaviors, but this research provides empirical support within a tourism context, demonstrating how tourists with positive environmental attitudes actively engage in sustainability practices, such as reducing waste, conserving resources, and choosing responsible tourism services ([45]).

Third, the study offers new insights into pull factors by showing that key tourist resources significantly shape environmental attitudes, whereas information, comfort, and accessibility do not. Previous literature has emphasized the role of destination attributes in sustainability engagement ([68]), but this research clarifies which pull factors effectively contribute to environmental attitude formation. The finding that sustainability information alone is insufficient to influence environmental attitudes aligns with studies suggesting that passive exposure to sustainability messages does not necessarily translate into behavioral change ([7]). Instead, tourists may require direct engagement with conservation initiatives to internalize environmental values.

Fourth, the study enhances the understanding of sustainable transportation and accessibility in tourism. While prior research has suggested that sustainable transport options influence travel behavior ([6]), this study shows that they do not necessarily alter environmental attitudes. Tourists may prioritize convenience, affordability, and efficiency over sustainability concerns when selecting transportation, indicating that structural improvements alone are insufficient to drive environmental attitude shifts ([35]). This underscores the need for integrated sustainability strategies that combine infrastructure development with educational programs and behavioral incentives to encourage more profound pro-environmental engagement. Moreover, while this study primarily focused on cognitive mechanisms through which push and pull motivations shape environmental attitudes, future research could adopt a dual-process perspective by integrating both cognitive and affective dimensions. Specifically, affective nature connectedness—the emotional sense of belonging and unity with the natural environment—may serve as a parallel mediator alongside cognitive attitude. Incorporating this affective pathway would deepen theoretical understanding by capturing the emotional aspects of tourists’ interactions with nature, which often drive sustainability-oriented behaviors beyond rational evaluation alone. Such a dual-process framework could provide a more holistic explanation of how both reasoning and emotion jointly shape eco-friendly intentions and actions.

### 5.2. Practical Contributions

Finally, this study has practical implications for tourism management and policy-making. The findings suggest that destination managers and policymakers should focus on fostering psychological engagement with nature, rather than solely providing sustainability information or improving eco-friendly infrastructure. For example, destination managers could implement nature-based storytelling sessions for children, host citizen science biodiversity counts, or offer family-oriented eco-restoration programs that foster emotional connection to local ecosystems and reinforce sustainability values. Since family-oriented experiences, appreciation of nature, and psychological restoration contribute significantly to environmental attitudes, sustainability initiatives should integrate experiential and emotional engagement strategies ([76]). For example, nature-based storytelling sessions, guided family eco-trails, or participatory conservation workshops could serve as targeted interventions to enhance emotional connection and sustainability awareness ([4]). Furthermore, tourism operators should emphasize eco-friendly activities that involve direct interaction with natural environments, as these experiences are more likely to reinforce pro-environmental attitudes and behaviors. It is important to distinguish ecotourism from sustainable tourism. Ecotourism primarily focuses on small-scale, nature-based, and educational travel experiences, whereas sustainable tourism encompasses a broader framework that integrates environmental, cultural, and socio-economic sustainability across all tourism forms. Clarifying this distinction enhances the interpretability and policy relevance of the findings. This conceptual clarification can strengthen the contextual validity of the study’s findings ([10]).

In summary, this study advances theoretical understanding by differentiating effective and ineffective push–pull factors, reinforcing the role of environmental attitudes in shaping sustainable behaviors, and highlighting the need for engagement-based sustainability strategies. Future research can build upon these findings by exploring additional psychological and contextual factors that further influence tourists’ environmental decision-making. Ultimately, the novelty of this study lies not in isolated statistical results but in its interdisciplinary integration of motivational theory and behavioral models within an East Asian sustainability context.

### 5.3. Limitations and Future Research Suggestions

Despite its contributions, this study has several limitations that should be acknowledged, which provide opportunities for future research. One limitation is the cross-sectional nature of the study, which captures tourist motivations, environmental attitudes, and eco-friendly behaviors at a single point in time. Since attitudes and behaviors may evolve over time, a longitudinal approach could offer deeper insights into how environmental attitudes develop and how they translate into long-term sustainable behaviors ([23]). Future research could employ longitudinal studies to examine how tourists’ motivations and behaviors change across multiple visits or after repeated exposure to sustainability initiatives.

Another limitation is the study’s reliance on self-reported data, which is subject to social desirability bias, meaning that participants may have overreported pro-environmental attitudes and behaviors to align with socially acceptable norms ([64]). To address this issue, future studies should consider observational methods or behavioral experiments to verify whether reported attitudes correspond to actual behaviors. Additionally, incorporating implicit attitude measures, such as the Implicit Association Test (IAT), could help reduce response bias and provide a more accurate understanding of tourists’ subconscious environmental attitudes.

The geographical scope of the study also presents a limitation, as findings are based on a specific destination context, which may limit their generalizability to other tourism markets or cultural settings. Since environmental attitudes and behaviors vary across different cultural backgrounds and regulatory environments, future studies should compare cross-cultural differences in tourist motivations and sustainability engagement. Expanding the research across diverse tourism settings—such as urban ecotourism, adventure tourism, and cultural heritage tourism—could provide a broader perspective on how push and pull factors influence environmental attitudes in different contexts. This study did not conduct robustness checks involving demographic control variables due to its exploratory design and theoretical focus. Future research may include such variables to assess whether the relationships remain stable across different demographic groups. Another limitation concerns the Adventure construct, which was measured using only two items with uneven factor loadings (0.971 vs. 0.759). Although the model met reliability thresholds, the restricted item coverage indicates limited content validity. Future research should expand this dimension to include broader sub-components such as risk-taking or personal challenge. Given this limitation, a sensitivity analysis excluding the Adventure factor could be conducted in future studies to examine whether structural relationships remain stable.

While this study differentiated between effective and ineffective push–pull factors, it did not consider the moderating role of personal values, environmental knowledge, or past experiences. In particular, variables such as prior environmental education, ecological worldview (i.e., New Ecological Paradigm, which measures an individual’s beliefs about human–nature relationships and ecological responsibility), or perceived destination quality (i.e., interpretation system efficacy, referring to how effectively a destination’s interpretive materials and programs communicate environmental or cultural meanings to visitors) may moderate or mediate the relationship between motivation and behavior, influencing the validity of causal inferences ([54]; [68]). Prior research suggests that personal environmental values strongly influence sustainable behaviors, meaning that individuals with deeply held ecological values may be more likely to engage in pro-environmental behaviors regardless of tourism motivations ([73]). Future studies should examine how psychological traits, past travel experiences, and environmental literacy moderate the relationships between motivation, attitude, and behavior.

Lastly, this study primarily focused on tourists’ environmental attitudes and behaviors, but future research should incorporate external factors, such as policy interventions, destination management strategies, and industry regulations, which could influence sustainability outcomes. Investigating how government policies, eco-certifications, and sustainable tourism practices interact with tourist motivations could provide practical insights for stakeholders aiming to design more effective sustainability programs ([9]).

In conclusion, while this study provides valuable insights into the influence of push and pull factors on environmental attitudes and eco-friendly behavior, future research should explore longitudinal changes, alternative measurement methods, cross-cultural perspectives, personal moderating factors, and policy interventions to develop a more comprehensive understanding of sustainability in tourism. Unlike prior studies that treat travel motivations as uniform predictors, this study unpacks their differential influence on environmental attitudes, offering a more nuanced framework for understanding eco-conscious travel behavior.

## Figures and Tables

**Figure 1 behavsci-15-01651-f001:**
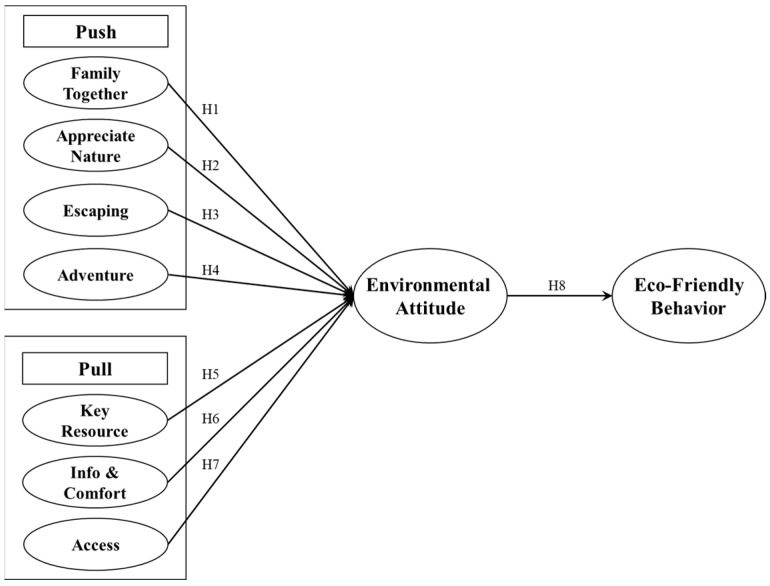
Theoretical Framework.

**Figure 2 behavsci-15-01651-f002:**
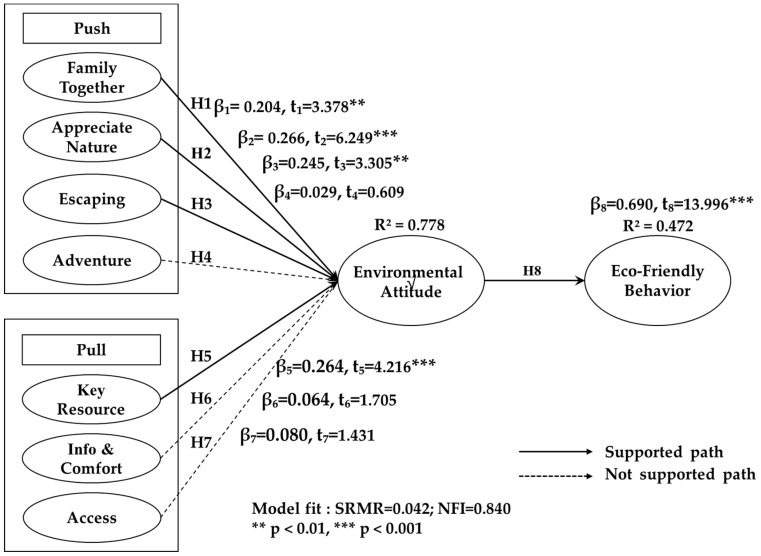
Path coefficients from PLSc results.

**Table 1 behavsci-15-01651-t001:** Reliability and convergent validity of measures.

Constructs	Indicators	Loadings	t-Values	Alpha	Rho_A	CR	AVE
Family Together				0.918	0.920	0.918	0.691
	To have time for natural study	0.865	27.749 ***				
	To have enjoyable time with family	0.791	22.489 ***				
	To observe rare wildlife	0.850	28.861 ***				
	To appreciate historic resources	0.885	27.602 ***				
	To experience cultural resources	0.759	20.847 ***				
Appreciating Nature				0.847	0.853	0.847	0.649
	To enjoy natural resources	0.720	16.239 ***				
	To enhance health	0.824	23.492 ***				
	To appreciate beautiful natural resources	0.866	21.601 ***				
Escaping				0.866	0.872	0.867	0.687
	To take a rest	0.894	38.478 ***				
	To get away from everyday life	0.757	21.100 ***				
	To avoid hot weather	0.830	29.020 ***				
Adventure				0.849	0.886	0.862	0.760
	To enjoy adventure	0.971	23.518 ***				
	To build friendship	0.759	18.680 ***				
Key Resources				0.885	0.886	0.885	0.606
	Rare fauna and flora (or aquatic plants/animals)	0.806	25.114 ***				
	Beautiful natural resources	0.732	19.574 ***				
	Tranquil rest areas	0.746	18.159 ***				
	Cultural and historic resources	0.814	22.054 ***				
	Well-conserved environment	0.791	22.912 ***				
Info & Comfort				0.908	0.915	0.908	0.713
	Well-organized tourist information system	0.825	11.739 ***				
	Convenient facilities (e.g., restroom, drinking stand)	0.727	8.216 ***				
	Convenient parking lots	0.936	13.920 ***				
	Clean and comfortable accommodations	0.875	10.748 ***				
Access				0.777	0.779	0.777	0.636
	Easy accessibility	0.820	24.489 ***				
	Convenient transportation	0.774	21.107 ***				
Environmental Attitude				0.913	0.915	0.914	0.602
	We are approaching the limit of the number of people the earth can support.	0.781	31.004 ***				
	The earth is like a spaceship with very limited room and resources.	0.802	32.256 ***				
	When humans interfere with nature it often produces disastrous consequences.	0.762	23.874 ***				
	The balance of nature is very delicate and easily upset.	0.762	22.610 ***				
	Humans are severely abusing the environment.	0.760	29.118 ***				
	The so-called “ecological crisis” facing humankind has been greatly exaggerated.	0.723	21.390 ***				
	If things continue on their present course, we will soon experience a major ecological catastrophe.	0.836	40.170 ***				
Eco-Friendly Behavior				0.931	0.933	0.931	0.692
	When I visit foreign countries as a tourist, I avoid buying goods with unnecessary packaging material.	0.814	25.848 ***				
	During my visit to foreign countries as a tourist, I buy environmentally friendly products whenever possible.	0.836	29.030 ***				
	I reduce and recycle waste, whenever possible, during my visits to foreign countries as a tourist.	0.888	31.874 ***				
	As a tourist, I always like to visit environmentally friendly countries.	0.874	30.923 ***				
	When I visit foreign countries as a tourist, I try to minimize my consumption of water and energy.	0.825	33.042 ***				
	When I visit foreign countries as a tourist, I choose means of transportation with the least ecological footprint.	0.836	19.778 ***				

Notes: ‘Alpha’ represents Cronbach’s alpha coefficient, while ‘Rho–A’ refers to Dijkstra–Henseler’s Rho–A coefficient. ‘CR’ stands for composite reliability, and ‘AVE’ indicates the average variance extracted. The symbol ‘***’ denotes a highly significant *p*-value, specifically at *p* < 0.001.

**Table 2 behavsci-15-01651-t002:** Discriminant validity of measures.

	A	B	C	D	E	F	G	H	I
A.FT	**0.832**	0.500	0.556	0.625	0.726	0.304	0.499	0.741	0.554
B.AN	0.502	**0.806**	0.389	0.400	0.519	0.102	0.396	0.648	0.606
C.ES	0.554	0.388	**0.829**	0.461	0.582	0.325	0.431	0.683	0.504
D.AD	0.620	0.399	0.453	**0.872**	0.535	0.300	0.477	0.575	0.411
E.KR	0.727	0.519	0.579	0.532	**0.779**	0.337	0.475	0.767	0.549
F.IC	0.306	0.105	0.326	0.295	0.339	**0.844**	0.245	0.351	0.085
G.AC	0.501	0.397	0.432	0.473	0.475	0.245	**0.798**	0.548	0.463
H.EA	0.743	0.648	0.684	0.570	0.768	0.354	0.549	**0.776**	0.685
I.EB	0.556	0.605	0.507	0.408	0.552	0.088	0.466	0.687	**0.832**

Notes: The bold values displayed along the diagonal correspond to the square root of the Average Variance Extracted (AVE). The lower triangular matrix presents the correlation coefficients between constructs, whereas the upper triangular matrix reflects the Heterotrait–Monotrait (HTMT) ratio.

## Data Availability

The original contributions presented in this study are included in the article. Further inquiries can be directed to the corresponding author.

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
