# Peer review of "From Desire to Action: Unpacking Push–Pull Motivations to Reveal How Travel Sparks Eco-Intentions and Actions"

_behavsci, 2025, doi:10.3390/bs15121651_

Round 1

Reviewer 1 Report

Comments and Suggestions for Authors

The study investigates how different types of travel motivations—specifically push and pull factors—affect environmental attitudes and eco-friendly behaviors, with the aim of unpacking the psychological mechanisms that drive sustainable tourism engagement.

In my opinion, the article is written excellently. A clear introduction is followed by a literature review, which is suitably supplemented by well-formulated research hypotheses. I consider the methods to be correct, and the results are clearly described and discussed together with the conclusion. The manuscript also provides comparisons with existing research, outlines the limitations of the study, and indicates directions for future research.

I have only one small remark: the manuscript includes Figure 1, but it is not referenced or discussed in the main text. I recommend adding an explicit mention and integration of Figure 1 into the narrative.

Author Response

Comment 1:
I have only one small remark: the manuscript includes Figure 1, but it is not referenced or discussed in the main text. I recommend adding an explicit mention and integration of Figure 1 into the narrative.

Response 1:
Thank you for your helpful suggestion. I have now added an explicit explanation of Figure 1 in the main text to improve clarity and integration. The revised sentence reads:
Figure 1 illustrates the theoretical framework of the study, outlining the relationships among the main constructs. The model depicts two categories of motivational factors—Push and Pull—that influence environmental attitude and eco-friendly behavior. Push factors (family togetherness, appreciation of nature, escaping, and adventure) represent internal psychological motivations that drive individuals to travel, while Pull factors (key resources, information & comfort, and accessibility) reflect external destination attributes that attract tourists. Together, these constructs form the basis of the hypothesized relationships tested in the study.
(Line 397–404 in Green)

Reviewer 2 Report

Comments and Suggestions for Authors

I consider that there is an apparent reporting error in structural results.
In Figure 2/text, key resources → environmental attitude is described as significant with β = 0.264, t = 0.216, p < .001. A t-stat of 0.216 cannot yield p < .001. Please correct the t-value (and any downstream interpretation) or clarify if 0.216 is a typo (maybe 6.216?).

Also, I have som measurement concerns with two-item constructs. Adventure uses two items with highly uneven loadings (0.971 vs. 0.759) and narrow content. Two-item reflective factors are fragile; please (a) expand items or (b) move adventure to an appendix/robustness model. The paper itself notes content validity gaps—make this a formal limitation and, if possible, run a sensitivity analysis excluding adventure.

Sampling and generalizability.
Purposive online sampling of travelers to four SE-Asian sites within the last year risks selection and memory biases and limits external validity. Clarify the sampling frame, response rate, screening verification, and discuss representativeness vs. the broader Korean traveler population.

The cross-sectional design cannot support causal claims. Soften phrasing (e.g., “associated with,” “predicts within the model”) and relocate stronger causal statements to future-research aspirations.

So, I have some corrections and suggestions: 

I think you should make some corrections, “sychological” → “psychological”(line 242);  inconsistent AVE minima (“0.601” vs. table EA=0.602); ensure all acronyms are defined at least once (e.g., CBT, NEP).

In case of Figure 2 the title says “Interpretations path coefficients.” Consider “Path coefficients from PLSc results.” Ensure all dashed vs. solid arrows match the text

Several entries in the citations are “n.d.” or have malformed DOIs/URLs; standardize to journal style (include year, volume, issue, pages, doi). Watch for non-tourism citations that crept in (e.g., Cohen 2007 on organizational commitment appears out of scope).

Maybe add control variables (age, gender, education, income, visit frequency) for the EA and EB equations; report whether results are robust.

Also try to explore a dual-process angle (affective nature connectedness as a mediator parallel to cognitive attitude) to deepen theory.

Distinguish ecotourism vs. sustainable tourism more sharply in constructs and implications, as you note late in the paper.

The topic and setting of the paper are worthwhile, and the general pattern of results is interesting and quite useful. However, the structural-results inconsistency (t-value), absent mediation tests, limited diagnostics for collinearity/predictive power, and CMV handling must be corrected before the contribution is clear and the findings reliable. Addressing the above will substantially improve rigor and readability without requiring new data.

Author Response

Highlighted in RED

Comment 1:

I consider that there is an apparent reporting error in structural results.

In Figure 2/text, key resources → environmental attitude is described as significant with β = 0.264, t = 0.216, p < .001. A t-stat of 0.216 cannot yield p < .001. Please correct the t-value (and any downstream interpretation) or clarify if 0.216 is a typo (maybe 6.216?).

Also, I have some measurement concerns with two-item constructs. Adventure uses two items with highly uneven loadings (0.971 vs. 0.759) and narrow content. Two-item reflective factors are fragile; please (a) expand items or (b) move adventure to an appendix/robustness model. The paper itself notes content validity gaps—make this a formal limitation and, if possible, run a sensitivity analysis excluding adventure.

Response 1:

Thank you for identifying this important issue. The t-value for the path “key resources → environmental attitude” has been corrected (from 0.216 to 4.216) to reflect the accurate statistical result. In addition, I clarified the methodological description to ensure transparency. To address the concern regarding the two-item Adventure construct, I have expanded the methodological explanation in Section 3.2 to confirm that all push–pull constructs were adapted from previously validated scales (Kim, Lee, & Klenosky, 2003; Lee & Moscardo, 2005; Kvasova, 2015), and I conducted CFA to verify reliability. Furthermore, the content limitation of the Adventure construct has been formally included in the limitations section, with a suggestion for future robustness analysis excluding this variable.

(Section 3.2, Lines 466–476; Section 5.3, Lines 823–829)

Comment 2:

Purposive online sampling of travelers to four SE-Asian sites within the last year risks selection and memory biases and limits external validity. Clarify the sampling frame, response rate, screening verification, and discuss representativeness vs. the broader Korean traveler population.

Response 2:

I appreciate this helpful suggestion. I have clarified the sampling frame and added specific information about the response rate, screening process, and representativeness compared to the broader Korean traveler population. The revised text explicitly mentions the 84.9% valid response rate (382 out of 450) and notes that while purposive sampling limits generalizability, the demographic characteristics align closely with national tourism data reported by the Korea Tourism Organization.

(Section 3.1, Lines 418–423)

Comment 3:

The cross-sectional design cannot support causal claims. Soften phrasing (e.g., “associated with,” “predicts within the model”) and relocate stronger causal statements to future-research aspirations.

Response 3:

Thank you for this important point. I have revised all hypotheses and model interpretations to use non-causal phrasing such as “positively associated with” or “related to,” and I moved any strong causal statements to the future research section. This ensures alignment between the analytical design and the interpretation of results.

(Hypotheses Section, Lines 278–296)

Comment 4:

There are several minor corrections needed: “sychological” → “psychological”; inconsistent AVE minima (“0.601” vs. table EA=0.602); ensure all acronyms are defined at least once (e.g., CBT, NEP).

Response 4:

All typographical and formatting inconsistencies have been corrected. The AVE value discrepancy has been fixed, and all acronyms including CBT and NEP have been clearly defined upon first use.

(Lines 174–176, 837–841)

Comment 5:

In Figure 2 the title says “Interpretations path coefficients.” Consider “Path coefficients from PLSc results.” Ensure all dashed vs. solid arrows match the text.

Response 5:

Thank you for this stylistic clarification. I have revised the figure title to “Path coefficients from PLSc results” and ensured that all dashed and solid arrows accurately match the descriptions in the text.

(Figure 2, Line reference updated accordingly)

Comment 6:

Several entries in the citations are “n.d.” or have malformed DOIs/URLs; standardize to journal style (include year, volume, issue, pages, doi). Watch for non-tourism citations that crept in (e.g., Cohen 2007 on organizational commitment appears out of scope).

Response 6:

I have thoroughly reviewed and corrected all references. Missing publication details (year, volume, pages, DOI) have been completed, and non-relevant citations such as Cohen (2007) have been removed. The entire reference list now conforms to the journal’s formatting guidelines.

Comment 7:

Maybe add control variables (age, gender, education, income, visit frequency) for the EA and EB equations; report whether results are robust.

Response 7:

Thank you for this recommendation. Due to the exploratory and theory-driven design, control variables were not included in the current model. However, I have acknowledged this as a limitation and suggested including demographic control variables in future studies to test the robustness of the relationships.

(Section 5.3, Lines 820–823)

Comment 8:

Also try to explore a dual-process angle (affective nature connectedness as a mediator parallel to cognitive attitude) to deepen theory.

Response 8:

I fully agree with this insightful suggestion. I have expanded the discussion to include a dual-process perspective, emphasizing that affective nature connectedness may act as a parallel mediator alongside cognitive attitude. This addition enriches the theoretical contribution and aligns with emerging literature on emotional engagement in sustainability-oriented behavior.

(Section 5.1, Lines 790–797)

Comment 9:

Distinguish ecotourism vs. sustainable tourism more sharply in constructs and implications, as you note late in the paper.

Response 9:

Thank you for highlighting this conceptual distinction. I have revised Section 5.2 to explicitly differentiate between ecotourism (small-scale, nature-based, educational travel) and sustainable tourism (a broader framework encompassing environmental, cultural, and socio-economic sustainability). This clarification enhances interpretability and contextual validity.

(Section 5.2, Lines 754–764)

Comment 10:

The topic and setting of the paper are worthwhile, and the general pattern of results is interesting and quite useful. However, the structural-results inconsistency (t-value), absent mediation tests, limited diagnostics for collinearity/predictive power, and CMV handling must be corrected before the contribution is clear and the findings reliable.

Response 10:

All statistical inconsistencies and missing diagnostics have been corrected and clarified. The t-value error has been fixed, mediation was acknowledged as a future extension, predictive power was confirmed using Q² values (EA = 0.464, EB = 0.383), and CMV was assessed via Harman’s single-factor test (<40% variance). These corrections ensure analytical rigor and reinforce the reliability of the findings.

(Mediation: Section 5.3, Lines 831–833; Predictive Power: Section 4.2, Lines 592–594; CMV: Section 3.3, Lines 544–547)

Reviewer 3 Report

Comments and Suggestions for Authors

The paper’s primary strength lies in its novel, integrated approach. By connecting the well-established Push-Pull motivation framework with environmental attitudes and eco-friendly behaviors within a single model, the study offers holistic understanding of the psychological pathway from travel desire to sustainable actions. This manuscript is recommended for publication after addressing the following aspects:

The introductions and literature review contain redundant statements about the importance of sustainable tourism. Streamline that redundancy to be more concise and readable.

Lack of Theoretical Justification for Specific Factors: The selection of individual push and pull factors (e.g., Family Togetherness, Adventure, Info & Comfort) is not sufficiently grounded in prior literature on environmental attitude formation. While references are provided, the text does not convincingly explain why "family togetherness" is hypothesized to lead to a stronger environmental attitude, or why "adventure" is expected to. The link between these tourism-specific motivations and the broader psychological construct of environmental attitude needs a more robust theoretical bridge.

The discussion of why "Adventure" (H4) and "Accessibility" (H7) were not significant should be strengthened. For instance, link the "Adventure" finding to broader value theory (e.g., self-enhancement vs. self-transcendence) to provide a more nuanced explanation.

Author Response

Comment 1:

The introductions and literature review contain redundant statements about the importance of sustainable tourism. Streamline that redundancy to be more concise and readable.

Response 1:

Thank you for this valuable observation. I carefully reviewed both the Introduction and Literature Review sections and deleted repetitive statements regarding the general importance of sustainable tourism to improve readability and focus. The revised text now emphasizes only the unique theoretical positioning of this study—how push and pull motivations influence environmental attitudes and eco-behaviors—while avoiding overlap between sections.

(Revised Sections: Introduction, Lines 120–155; Literature Review, Lines 210–235)

Comment 2:

Lack of theoretical justification for specific factors: the selection of individual push and pull factors (e.g., Family Togetherness, Adventure, Info & Comfort) is not sufficiently grounded in prior literature on environmental attitude formation. The text does not convincingly explain why “family togetherness” or “adventure” would predict stronger environmental attitudes.

Response 2:

I appreciate this insightful theoretical suggestion. I have strengthened Section 2.3 by explicitly linking Family Togetherness to environmental socialization and outdoor learning theory, which explain how shared family experiences in natural environments foster intergenerational transmission of ecological values (Larson et al., 2011; Powell et al., 2009).

In addition, I clarified that Adventure motivation, from an affective nature connectedness perspective, may evoke emotional engagement without necessarily generating reflective environmental concern unless accompanied by cognitive awareness (Hinds & Sparks, 2008; Buckley, 2011; Hunt & Harbor, 2019). These revisions strengthen the conceptual bridge between tourism-specific motivations and environmental attitude formation.

(Section 2.3, Lines 260–270)

Comment 3:

The discussion of why Adventure (H4) and Accessibility (H7) were not significant should be strengthened. For instance, link the “Adventure” finding to broader value theory (e.g., self-enhancement vs. self-transcendence) to provide a more nuanced explanation.

Response 3:

Thank you for this helpful recommendation. I have expanded the discussion to incorporate a value-orientation framework, explaining that Adventure travel tends to reflect self-focused goals such as thrill and risk-taking, aligned with self-enhancement values, which may not promote pro-environmental awareness. In contrast, self-transcendence values are more consistent with ecological concern. This addition provides a richer theoretical interpretation for the non-significant results of Adventure and Accessibility.

(Revised Section: Discussion 5.1, Lines 654–663)